# Cistanche promotes the adipogenesis of 3T3-L1 preadipocytes

Ping Zhang[1☯], Le Su[1☯], Xiuyu Ji[1], Feifan Ma[1], Qiulin Yue[1], Chen Zhao[2], Song Zhang[1], Xin Sun[1], Kunlun Li[3], Lin Zhao[1,3]*

1 State Key Laboratory of Biobased Material and Green Papermaking, School of Bioengineering, Shandong Academy of Sciences, Qilu University of Technology, Jinan, China, 2 Shandong Provincial Key Laboratory of Food and Fermentation Engineering, Shandong Food Ferment Industry Research & Design Institute, Shandong Academy of Sciences, Qilu University of Technology, Jinan, China, 3 Jinan Hang Chen Biotechnology Co., Ltd., Jinan, China

☯ These authors contributed equally to this work.
* iahb205@163.com

**Data Availability Statement:** RNA sequencing data is available at NCBI Sequence Read Archive (SRA) and the relevant accession submission numbers are: SUB10721702. The BioProject ID is PRJNA783654.

## Abstract

*Cistanche deserticola Ma* (cistanche) is a traditional herb with a wide range of therapeutic properties. However, no evidence of cistanche's effect on adipogenesis has been found. The effect of cistanche that promotes the adipogenesis of 3T3-L1 preadipocytes was proved by using MTT spectrophotometry, Nile Red staining, Oil Red O staining and transcriptome sequencing technology. The mRNA level of key transcription factors for adipogenesis such as PPAR, AP2 and LPL were examined by RT-PCR. The results showed that the intracellular lipid content in cistanche treated cells were notably increased when compared with the non-treated cells. Between the differentiation and cistanche treated groups, the expression of adipogenesis related genes such as grow hormone releasing hormone (Ghrp), BCL2/adenovirus E1B interacting protein 3 (Bnip3) and Gastric inhibitory polypeptide receptor (Gipr) were significantly increased. Our findings also verified that cistanche promoted adipogenesis, which was accompanied by up-regulated level of Bnip3 and PPAR. This study could uncover new signaling pathways involved in adipogenesis regulation.

## Introduction

Adipocytes are a key component of adipose tissue, and their proliferation and differentiation are linked to the tissue's function. Adipocyte metabolism is abnormal in a range of disorders, including obesity, nutritional insufficiency and diabetes [1]. The study of adipocytes in vitro has become a popular issue in recent years due to the rising prevalence of lipid metabolism disorders. The 3T3-L1 preadipocytes are one of the most often used cells to create adipocytes models, and under the right conditions, they can be differentiated into adipocyte cells [2]. Knowing the key factors and signaling pathways in the process of adipogenesis could lead to a more effective strategy for treating lipid metabolism problems. As a result, it is vital to investigate the critical components and signaling pathways involved 3T3-L1 preadipocyte adipogenesis.

**Funding:** This work was supported by Spring Industry Leader Talent Support Plan (grant number 2017035 and 2019042), Shandong Tai shan leading talent project (grant number LJNY202015), Key R&D Program of Shandong Province (grant number 2019YYSP019 and2019QYTPY024); Science, education and industry integration and innovation pilot project of Qilu University of Technology (Shandong Academy of Sciences) (grant number2020KJC-YJ01 and 2020KJC-GH10), National key plan "science and Technology to help the economy" special project and University, government, industry, research Collaborative innovation Fund project (2020-CXY45). Yantai Development Zone Science and Technology Leading Talents Project (grant number 2020CXRC4).

**Competing interests:** The authors have declared that no competing interests exist.

Cistanche is a traditional Chinese medication that has been used to cure disease and maintain health for thousands of years [3]. Cistanche has been shown to reduce the level of malondialdehyde (MDA), which is formed during the lipid peroxidation process [4, 5]. And lipid peroxidation is the process of oxidation in lipids and finally results in cell damage [6]. Furthermore, a range of natural compounds in cistanche have been shown to affect lipid metabolism by regulating triglyceride (TG) and fatty acid levels [5, 7, 8]. Moreover, cistanche has been shown to stimulate the differentiation of bone mesenchymal stem cells (BMSCs) into osteoblasts [9]. Cistanche promoted the development of murine marrow-derived dendritic cells (BMDCs) in a considerable way [10]. Therefore, we postulated that cistanche could have the potential capacity to promote the differentiation of 3T3-L1 preadipocytes.

RNA sequencing (RNA-Seq) has become an effective method for large scale transcriptomic research. RNA-Seq was able to extract nearly all complementary sequences transcripts (cDNA) from cell RNA [11]. DEGs were analyzed using GO enrichment to analyze the biological processes, cellular components and molecular function in DEGs. The DEGs enrichment signaling pathway was investigated using the Kyoto Encyclopedia of Genes and Genomes (KEGG) Pathway [12]. Thus, specific components implicated in adipogenesis of cistanche treated 3T3-L1 cells could be obtained in our study using RNA-Seq technology [13, 14].

## Materials and methods

### Chemicals and reagents

Shandong Institute of Traditional Chinese Medicine was where Cistanche was obtained. Solebol Technology Co. LTD provided acid cellulose and acid pectinase. Nile red and Oil red O were purchased from Sigma (St. Louis, MO, USA). Dulbecco's Modified Eagle's Medium-Hight glucose (DMEM-H) and fetal bovine serum (FBS) were obtained from Gibco, lnc (Grand Island, NY). The following items were acquired from Sigma: isobutylmethylxanthine (IBMX), dexamethasone, insulin and 3-(4,5-dimethylthiazol-2-yl)-2,5-diphenyltertrazolium bromide(MTT)(St. Louis, Mo).

### Preparation of cistanche

All plant experiments complied with the IUCN Policy Statement on Research Involving Species at Risk of Extinction and the Convention on the Trade in Endangered Species of Wild Fauna and Flora. The cistanche was divided into three different treatments: Boiled, Enzymed and Boiled with Enzymed. (1) *Boiled* ultramicro mill was used to crush cistanche into powder with a mesh greater 1500 mesh. The cistanche powder was dissolved in distilled water at a ratio of 1:9. After uniform stirring, the solution of cistanche was extracted by boiling at 100˚C for 30 minutes. The supernatant was saved and suspended in a tenfold ethanol solution before being heat refluxed three times for 30minutes at 100˚C. The sample was centrifuged at 10000 r/min and the supernatant was lyophilized for preservation. (2) *Enzymed* the cistanche was ultrafinely pulverized and dissolved in sterile water before being enzymed at 50˚C for 2 hours with acid pectinase and acid cellulase (acid fiber and sour fruit amount accounted for 1 ‰ of solid). The sample was centrifuged at 10000 r/min and the supernatant was lyophilized for preservation. (3) *Boiled with Enzymed* after pulverized cistanche solution boiling at 100˚C for 30 minutes, the boiled cistanche extraction was then done at 50˚C for 2 hours by adding acid pectinase and acid cellulose. The supernatant was then saved and suspended in a tenfold ethanol solution, and the heat was refluxed three times for 30 minutes at 100˚C. The sample was centrifuged at 10000 r/min and the supernatant was lyophilized for preservation.

## 3T3-L1 preadipocytes culture and differentiation

3T3-L1 preadipocytes (presented by professor Yanqing Li from Qilu hospital of China) were cultured in Dulbecco's Modified Eagle's Medium-Hight glucose (DMEM-H, Gibco, 12800–017) supplemented with 10% fetal bovine serum (FBS; v/v) (Hyclone, SV30087.02). Cells were kept at 37°C in a humidified atmosphere of 5% $CO_2$ under standard conditions.

The 3T3-L1 preadipocytes were divided into 5 groups: the normal control group (Nor), the differentiation group (Differ), the boiled group (Boiled), the enzymed group (Enzymed) and the boiled with enzymed group (Boiled with Enzymed). The Boiled group, Enzymed group and Boiled with Enzymed group were respectively added with 0.5μg/ml of cistanche extraction into the induction medium [15].

3T3-L1 cells were seeded at a density of $1\times10^5$ cells/ml in a 6-well culture plate. Two days after cells reached confluence, Cell differentiation was then induced with 10% FBS DMEM contain MDI (0.5μM isobutylmethylxanthine IBMX, 5μM dexamethasone and 0.5 μg/ml insulin) with cistanche for 2 days. Next, the medium was replaced with 10% FBS DMEM containing 5 μg/ml insulin with cistanche for 2 days. Next, the differentiation medium was replaced with 10% FBS DMEM contains cistanche for another 2 days. On day 7 of the experiment, Oil red O and Nile Red staining was performed, and the triglyceride, lipoprotein and lipid of intracellular could be stained into jacinth and red by ORO and Nile Red staining.

## Cell viability assay

3-[4,5-dimethylthiazol-2-yl -2,5-diphenyltertrazolium bromide (MTT) assay was used to assess cell viability in response to cistanche. Briefly, 3T3-L1 preadipocytes were seeded in 96-well plates at a density of $5 \times 10^4$ cells/100 μl and treated with 0.05, 0.5 and 1μg/ml concentration of different cistanche extraction. After 24h and 48h, the culture media was added 20 μl MTT reagent to each well for another 4 h at 37°C. The absorbance was measured using a SpectraMax ABS microplate spectrophotometer (Molecular Devices, USA).

## Oil red O staining

3T3-L1 preadipocytes were washed with PBS and fixed in 10% formaldehyde in PBS for 1 hour. Then cells were washed with 60% isopropanol and stained with working solution oil red for 10 minutes. The stained cells were washed four times with double distilled water and photographed under LSM 510 inverted microscopy. In order to determination the lipid content in 3T3-L1 adipocytes, cells were dissolved in isopropanol and the absorbance values of eluates were measured using a SpectraMax ABS microplate spectrophotometer (Molecular Devices, USA).

## Nile red staining

3T3-L1 preadipocytes were fixed with 4% paraformaldehyde for 20 minutes, and then the cells were stained with NR for 10 minutes. The stained cells were washed three times with phosphate-buffered saline (PBS). The images of stained lipid droplets were taken using fluorescence microscope BZ-ZX700 (Keyence, Osaka, Japan). To quantify the lipid content, cells were dried and isopropanol was added, Absorbance values of eluates were measured using a SpectraMax ABS microplate spectrophotometer (Molecular Devices, USA) at 520 nm wavelength [16].

## Quantitative real-time RT-PCR (RT-qPCR)

Total RNA was extracted from 3T3-L1 preadipocytes by Trizol (Invitrogen, USA), and RNA was reverse transcribed into cDNA using an ABScript II RT Mix (ABclonal, Wuhan, China).

The cDNAs were subjected to qRT-PCR using primer pairs 5′–AGATCATTTACACAATGC TGGC–3′ and 5′–TAAAGTCACCAAAAGGCTTTCG–3′ for PPAR, 5′–CATCCGGTCAGAGAG TACTTTT–3′ and 5′– TAGGGTTATGATGCTCTTCACC–3′ for AP2, 5′–CCTGATGACGCT GATTTTGTAG–3′ and 5′–CAATGAAGAGATGAATGGAGCG–3′ for LPL, 5′–CCACTAACGAA CCAAGTCAGAC–3′ and 5′–CATCTCTGCTGCTCTCTCAT–3′ for Bnip3, 5′–TGTGTCCGT CGTGGATCTGA–3′ and 5′–TTGCTGTTGAAGTCGCAGGAG–3′ for GAPDH. (Sangon Biotech Co., Shanghai, China). The mRNA expressions levels were examined on Rotor-Gene Q instrument (QIAGEN, Shanghai, China) using a SYBR Green Fast qPCR Mix (ABclonal, Wuhan, China) to evaluate the amount of double stranded DNA. The purity and concentration of isolated RNA was determined using NanoDrop 1000 (NanoDrop Technologies, Wilmington, DE, USA). The primer amplification efficiencies were measured in cDNA dilutions from 10 to $10^5$. And the amplification was linear over the range of 10 to $10^5$. The standard curves efficiencies of the primers ranged from 97% to 102%. The thermal cycling included 1 cycle at 95˚C for 30 s, 40 cycles at 58˚C for 30 s, at 72˚C for 30 s, and the final cycle at 72˚C for 5min. LPL, AP2, PPAR and Bnip3 gene expression levels were calculated using the normalized relative quantification method followed by the ΔΔCT method.

## Transcriptome profiling by RNA-Seq

Total RNA was isolated from 3T3-L1 preadipocytes using the TRIzol reagent (Invitrogen, Beijing, China) according to the manufacturer protocols. Equal amounts of total RNA samples were collected from 5 groups: the Nor group, Differ group, Boiled group, Enzymed group and Boiled with Enzymed group. The RNA samples were extracted from each group and different parallel set was obtained. Transcriptome sequence (RNA-Seq) was performed by HuaDa Gene Company.

## Statistical analysis

Data were analyzed using Excel and Graph Pad Prism and presented as mean ± SEM. Differences between Nor, Differ, Boiled, Enzymed or Boiled with Enzymed groups were analyzed by one-way analysis of variance (ANOVA) with use of SPSS v11.5 (SPSS Inc., Chicago, IL). P values less than 0.05 were considered as significant.

## Results

### Effect of cistanche extraction on 3T3-L1 preadipocytes cell viability

To explore the cytotoxic effects of cistanche extraction on 3T3-L1 preadipocytes, cell viability was measured by using the MTT assay (Fig 1A). Our results showed that cistanche at 0.5μg/ml had no cytotoxic effect on 3T3-L1 preadipocytes after 24h and 48h treatments (*P<0.05, P<0.01, P<0.001*). Therefore, the concentration of cistanche at 0.5μg/ml was used in the Boiled, Enzymed and Boiled with Enzymed groups in the next research. The 3T3-L1 preadipocytes could be differentiated in induction medium containing FBS, IBMX, dexamethasone and insulin. The results indicated that the induction medium in differ group could increase the viability of 3T3-L1 preadipocytes by 7 day. Meanwhile, the cell viability in the Boiled group also showed a significant increase compared to that in the Nor group. However, the cell viability exhibited no significant differences between differ and cistanche-treated groups (Fig 1B) (*P<0.05, P<0.01*).

### Cistanche increased lipid accumulation in 3T3-L1 preadipocytes cells

The ORO and NR stainings were used to detect the intracellular lipid droplets in 3T3-L1 preadipocytes treated with cistanche extraction to investigate the influence of cistanche on

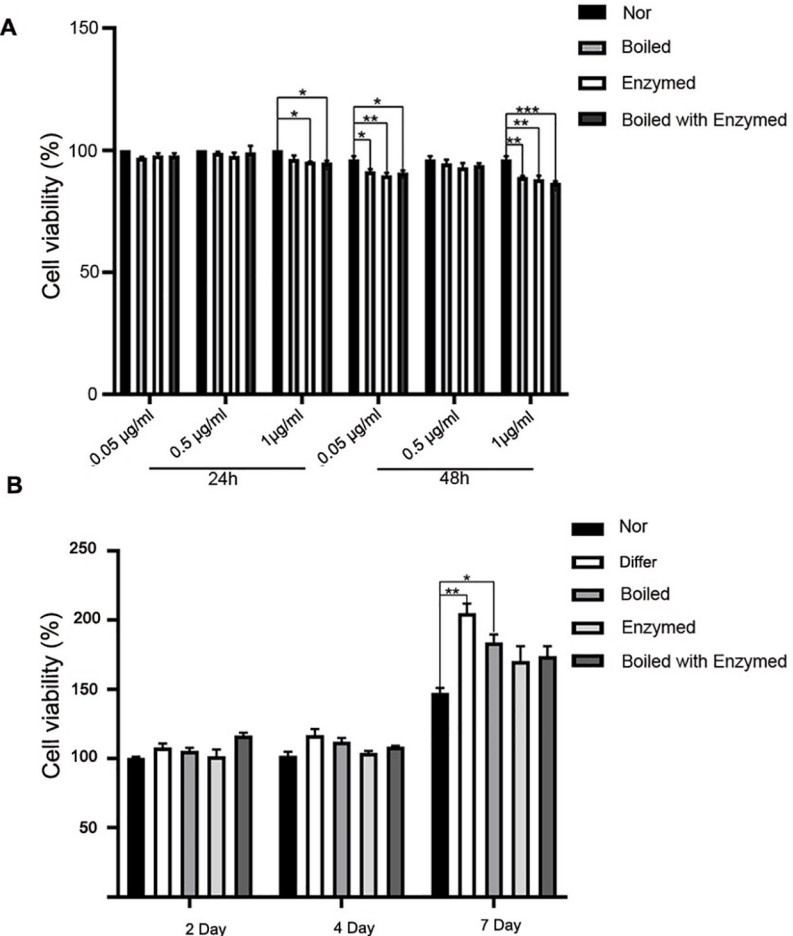

**Fig 1. The cell viability and cell proliferation in 3T3-L1cells after treated with cistanche.** (A) 3T3-L1 preadipocyte cells were treated with boil, enzyme and boil with enzyme of cistanche for 24h and 48h. (B) 3T3-L1 preadipocyte cells were pretreated in differentiation medium with cistanche for 7days. (Data are expressed as means ± S.E. $^*p < 0.05$, $^{**}p < 0.01$, $^{***}p < 0.001$. Experiments were performed in three repetitions).

adipogenesis. The intracellular lipid deposition was significantly increased in the Differ, Boiled and Boiled with Enzymed groups, according to the results of ORO staining (Fig 2A). In comparison to the Differ group, the levels of lipid accumulation were significantly increased in Boiled and Boiled with Enzymed group (Fig 2A–2f) (*P<0.01, P<0.001*). According to the results of NR staining, the lipid droplets are demonstrated in red color. The stained lipid droplets were dissolved in 100% isopropanol and the absorbance at 520nm was measured to determine their quantification. The accumulation of intracellular lipids was significantly increased in the group of Boiled and Boiled with Enzymed (Fig 2B). These results suggested that the cistanche extraction could promote the accumulation of intracellular lipids, and the cistanche of Boiled group and Boiled with Enzymed group exhibited better effects (*P<0.01, P<0.001*).

## RNA-seq analysis

RNA-seq analysis was performed to explore the mechanism of cistanche promotion the accumulation of intracellular lipids in 3T3-L1 preadipocyte cells. We calculated the pairwise Pearson correlation coefficients of FPKM values between samples. And the heat map was produced

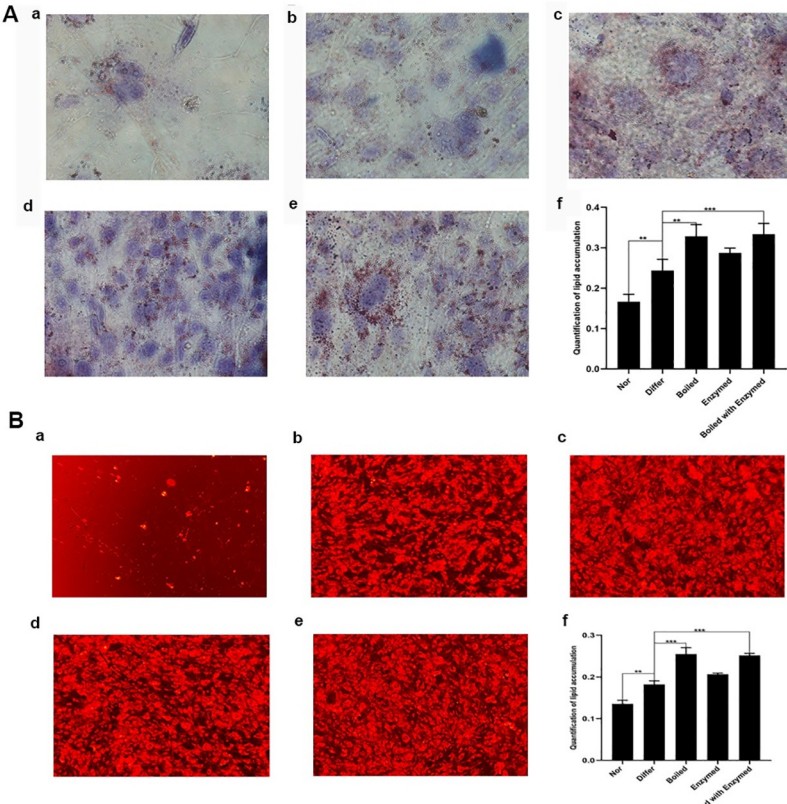

**Fig 2. Representative images of ORO-stained and Nile Red-stained 3T3-L1 cells differentiated.** (**A**) ORO staining of differentiated 3T3-L1 adipocytes. (a) Normal group. (b) 3T3-L1 cells were treated with differentiation medium. (c) 3T3-L1 cells were treated with cistanche of Boiled. (d) 3T3-L1 cells were treated with cistanche of Enzymed. (e) 3T3-L1 cells were treated with cistanche of Boiled with Enzymed. (f) Lipid accumulation was quantified by measuring the absorbance at 520 nm. (Data are expressed as means ± S.E.) The lipid drops was stained in red and nucleus was stained in blue, and the more red dots in the figure, the more lipid accumulation (**B**) NR staining of differentiated 3T3-L1 adipocytes. The lipid droplets are demonstrated in red color (a) Normal group. (b) 3T3-L1 cells were treated with differentiation medium. (c) 3T3-L1 cells were treated with cistanche of Boiled. (d) 3T3-L1 cells were treated with cistanche of Enzymed. (e) 3T3-L1 cells were treated with cistanche of Boiled with Enzymed. (f) Lipid accumulation was quantified by measuring the absorbance at 520 nm. (Data are expressed as means ± S.E. $^{**}p < 0.01$, $^{***}p < 0.001$. Experiments were performed in three repetitions).

[17]. The results showed that these samples were highly correlated and did not have larger difference in the same group. The experimental groups were separated from control group [18] (Fig 3).

As shown in the Venn diagram (Fig 4), 188, 231, 251, 220 and 233 genes were respectively detected in Nor, Differ, Boiled, Enzymed and Boiled with Enzymed groups (Fig 4A). A total of 14251 co-expressed genes were found between the Differ and experimental groups (Fig 4B). 368, 11 and 44 differential genes were respectively detected between Differ versus Boiled group, Differ versus Enzymed group, Differ versus Boiled with Enzymed group (Fig 4C). Among these detected differential gens, 9 genes were differentially expressed among Differ and experimental groups.

These differentially expressed genes were shown in the volcano plot, which the red colors represented the up-regulated genes, and the green colors represented the down-regulated genes (Fig 5). Comparison of 368 diff-expressed genes revealed 312 genes up-regulated and 56 genes down-regulated by setting the significance and fold-change (P<0.05 and log2 fold

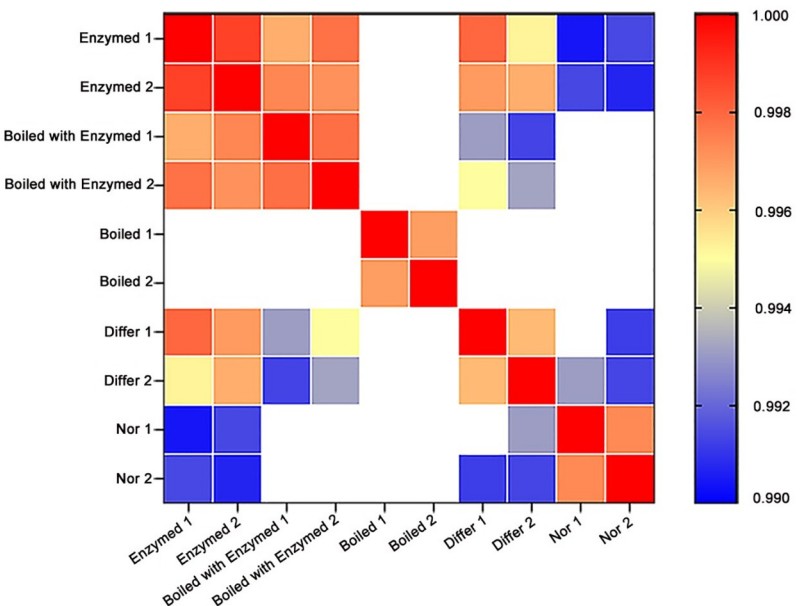

**Fig 3. The Pearson correlation coefficients of all gene expressions.** Pearson correlation coefficients were used to describe correlations between control and experimental group. The Pearson correlation coefficients of all gene expressions between each two samples were compared. The higher the correlation coefficient was, the more similar the gene expression level was. And the correlation coefficient values above 0.995 are presented as red and yellow, the correlation coefficient values above below 0.995 are indicated in blue. These samples were highly correlated and did not have larger difference in the same group, and cistanche experimental group were separated from control group.

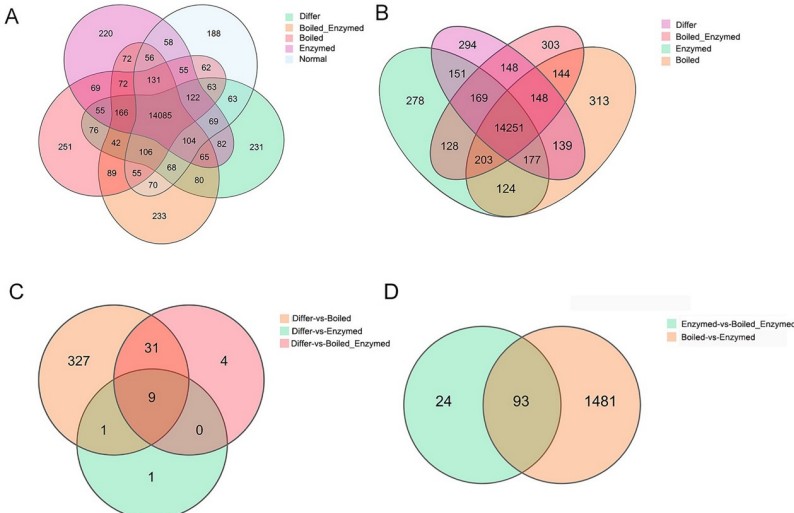

**Fig 4. Venn diagram shows the overlap of control and experimental group.** (A) 188, 231, 251, 220 and 233 genes were respectively detected in Nor, Differ, Boiled, Enzymed and Boiled with Enzymed group. (B) 14251 co-expressed genes were found between the Differ and experimental groups. (C) 368, 11 and 44 differential genes were respectively detected between Differ versus Boiled group, Differ versus Enzymed group, Differ versus Boiled with Enzymed group.

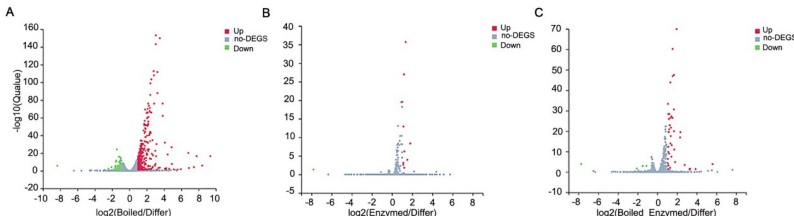

**Fig 5. Volcano plot shows the up or down regulated genes between control and experimental group.** (a) 312 genes up-regulated and 56 genes down-regulated between Boiled versus Differ. (b) 10 genes up-regulated and 1 gene down-regulated between Enzymed versus Differ. (c) 40 genes up-regulated and 4 genes down-regulated between Boiled with Enzymed versus Differ. ($P<0.05$ and log2 fold change>|1|).

change>|1|) (Fig 5A). Comparison of 11 diff-expressed genes revealed 10 genes up-regulated and 1 genes down-regulated by setting the significance and fold-change ($P<0.05$ and log2 fold change >|1|) (Fig 5B). Comparison of 44 diff-expressed genes revealed 40 genes up-regulated and 4 genes down-regulated by setting the significance and fold-change ($P<0.05$ and log2 fold change >|1|) (Fig 5C).

## Enrichment analysis of the Gene Ontology (GO) terms of the DEGs functional

In order to analysis the biological relationship of DEGs in adipocytes of cistanche treated cells. We performed GO enrichment analysis of DEGs. The GO enrichment histogram directly showed the quantity distribution of different expressed genes in biological processes, cellular components and molecular function (Fig 6). Within the biological processes category, most DEGs were associated with cellular process and metabolic process. In the cellular component category, the majority of DEGs were related with cell part and organelle part. In the molecular function category, most DEGs were associated with catalytic activity and transcription regulator activity.

## Kyoto Encyclopedia of Genes and Genomes (KEGG) pathway analysis of DEGs

The top 20 significant pathways of DEGs were selected and represented in KEGG enrichment scatter diagram (Fig 7). DEGs were mainly enriched in metabolic pathways, the Biosynthesis of secondary metabolites pathway and the Galactose metabolism pathway [19]. On the other hand, the down regulated genes were mainly enriched in the tumor necrosis factor (TNF) signaling pathway [20], forkhead box O (FoxO) signaling pathway [21], and other fundamental biochemical process, such as Glycolysis pathway [22], Biosynthesis of amino acids signaling pathway and Glucagon signaling pathway [23].

## Analysis of key transcription factors for adipogenesis

To validate changes in gene expression patterns, we examined the effect of cistanche on mRNA expression of important transcription factors such as AP2, LPL and PPAR, which are considered as adipogenic markers. The data showed that the levels of AP2, LPL and PPAR were significantly increased in Differ group. Compared with those in Differ group, the level of PPAR was significantly up-regulated in Boiled group. And in the Boiled with Enzymed group, levels of PPAR, AP2 and LPL were significantly higher than that in the Differ group (Fig 8)

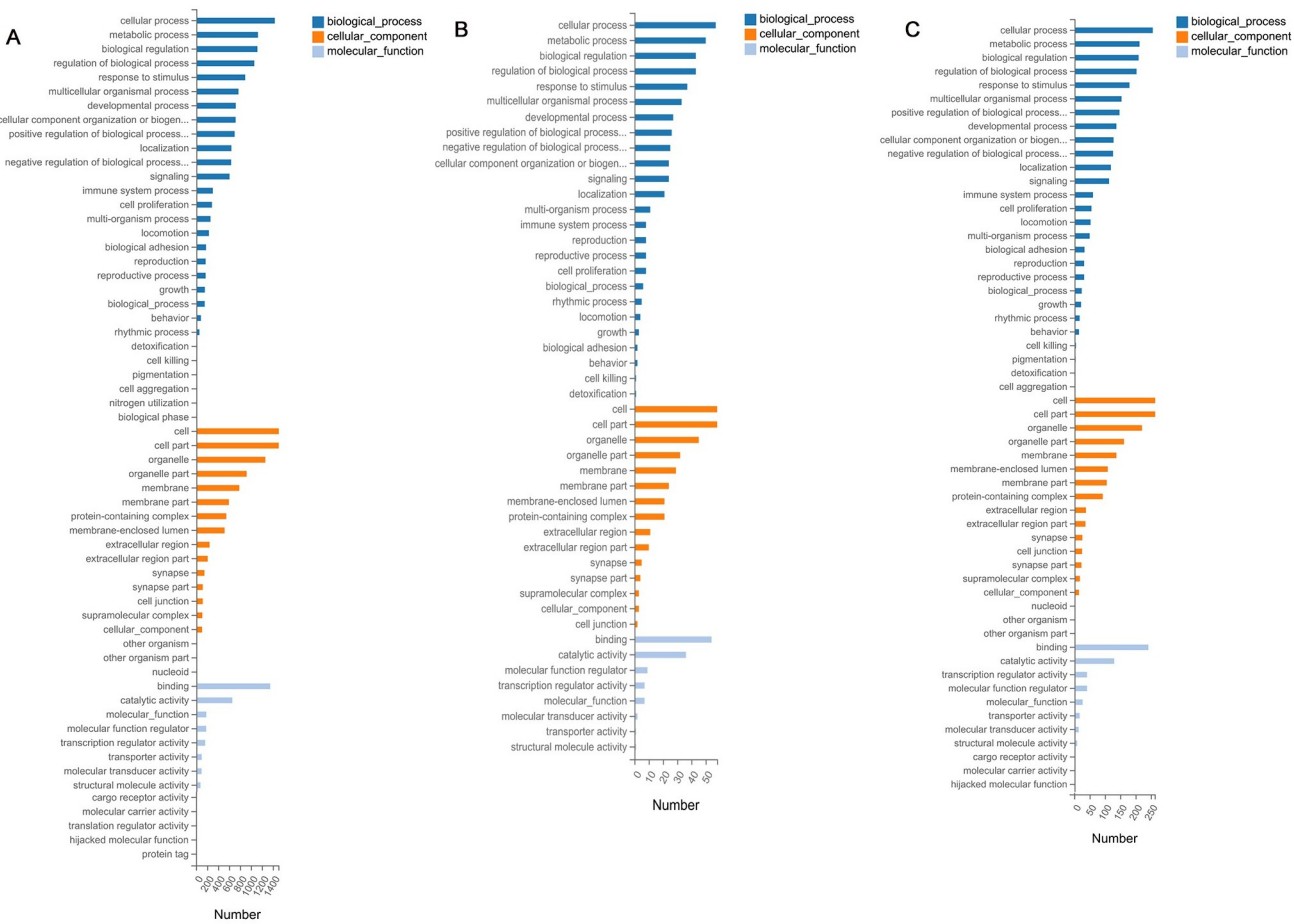

**Fig 6. Go analysis of DEGs.** The figure is consisting of molecular functions, cellular components and biological process. The significance level of enrichment was set at corrected *P*-value. (*P*<0.05 log2 fold change>|1|) (A) Differ versus Boiled group. (B) Differ versus Enzymed group. (C) Differ versus Boiled with Enzymed group.

(*P<0.01*). These experimental results suggested that the mRNA levels of AP2, LPL and PPAR were increased in ciatanche of Boiled with Enzymed group.

### Identify and verify the key genes in the regulation of adipogenesis

In order to identify the candidate key genes that potentially regulates adipogenesis of 3T3-L1 cells with cistanche [24], we compared the DEGs expression changes between control and experimental group. In comparison the Nor versus Differ group and Differ versus experimental group, the mRNA level of Bnip3 gene was all significantly increased (S1 Fig). And the mRNA level of Bnip3 also up-regulated in Boiled group, which is in line with the results of RNA-sequences (Fig 8D) (*P<0.05*). The results suggested that the accumulation of lipid in cistanche treated 3T3-L1 cell might be related to Bnip3.

### Discussion

Cistanche is a very valuable herb in Chinese traditional medicine [25]. Until now, a large number of biologically active substances in cistanche have been proved [6]. The polysaccharides of cistanche could modulate the lipid metabolism by regulating triglyceride (TG) and fatty acid

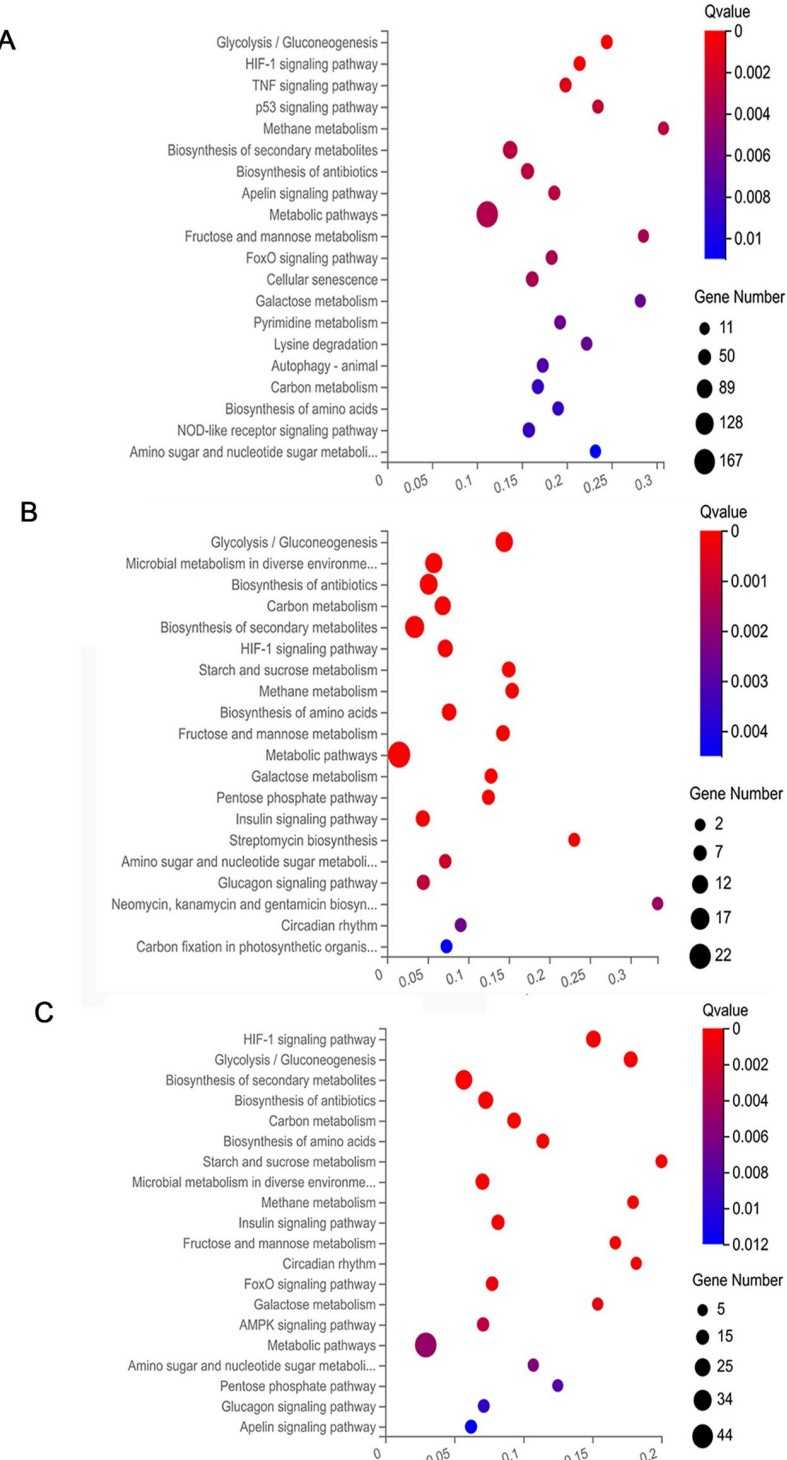

**Fig 7. KEGG pathway analysis of DEGs between Differ, Boiled, Enzymed, Boiled with Enzymed group.** The figure shows top 20 pathways in a total number of enrichment of DEGs in signaling pathway. Size and color of bubble represent amount of DEGs enriched in pathway and enrichment significance. (A) Differ versus Boiled group. (B) Differ versus Enzymed group. (C) Differ versus Boiled with Enzymed group. (*P*<0.05 and log2 fold change>|1|).

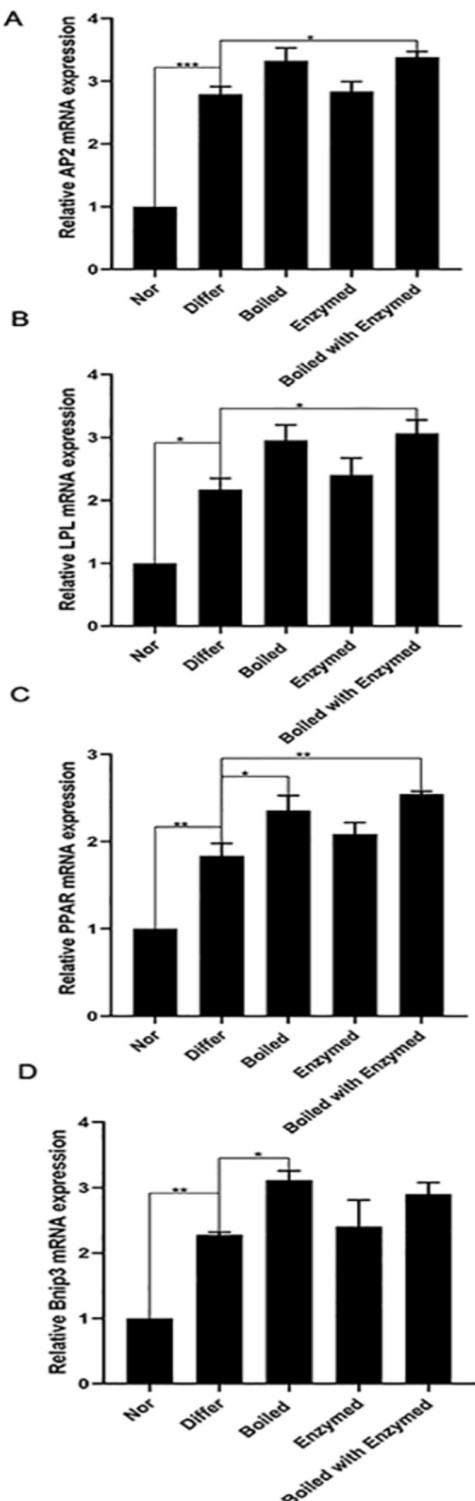

**Fig 8. Analysis of key transcription factors.** The mRNA levels of three key transcription factors AP2(A), LPL(B), and PPAR(C) Bnip3(D) in 3T3-L1 adipocytes were detected. Data are expressed as means ± S.E. $^*p < 0.05$, $^{**}p < 0.01$ and $^{***}p < 0.001$. Nor: Normal group. Differ: 3T3-L1 cells were treated with differentiation medium. Boiled: 3T3-L1 cells were treated with cistanche of Boiled. Enzymed: 3T3-L1 cells were treated with cistanche of Enzymed. Boiled with Enzyme: 3T3-L1 cells were treated with cistanche of Boiled with enzymed.

[4]. And the echinacoside of PhGs could affect the leptin biosynthesis pathway by regulating the gene of SOCS-3 [5]. Therefore, we indicated that cistance has an effect on adipogenesis, and it was selected to explore the accumulation of lipid in 3T3-L1 preadipocyte. Previous studies has showed that cistanche has a wide range of possible effects on lipid peroxidation and lipid accumulation, but its regulatory effects and mechanism on 3T3-L1 adipocytes are unknown. We expounded the mechanism of cistanche promoted the adipogenesis from the droplet formation and genetic level, which may help bridge this gap. Moreover, our present study also demonstrated that the levels of key transcription factors and DEGs were significantly increased in cistanche treated group, which help us to understanding the various adipocyte specific genes in the process of adipogenesis. This research has advanced our understanding of adipogenesis mechanisms while also demonstrating a potential target for adipogenesis regulation.

The process of adipogenesis is accompanied by altered expression of various transcription factors and adipocyte specific genes. Acid-binding protein (AP2), and Lipoprotein lipase (LPL) and Peroxisome proliferator activated receptor gamma (PPAR) are the key transcription factors in the process of fat synthesis. To further characterize the effect of cistanche on adipogenesis, we examined the expression of these key transcription factors using quantitative real-time PCR. We found that the expression of AP2, LPL and PPAR were up-regulated in differ group, which is consistent with the previous results. It was showed that the mRNA levels of these key transcription factors were also increased in cistanche treated cells, and the cistanche of Boiled with Enzymed group exhibited better ($P<0.05$, $P<0.01$), which to verify the results of ORO and NR stainings. It is might be that cistanche could be decomposed thoroughly by Boiled and enzyme, and the active ingredients of cistanche could be fully promoted in the group of Boiled with Enzymed. Therefore, the selection of extraction method is very important in future investigation of cistanche, and different extraction method may play different functional effects.

Meanwhile, the RNA-seq technology was used to explore the DEGs and gene expression patterns in control and cistanche treated 3T3-L1 cells. In additionally, our results showed that the mRNA levels of Bnip3 were up-regulated in cistanche treated group. A relatively studies have reported that Bnip3 encodes the mitophagy signaling pathway, which is associated with the lipid metabolism in the liver [26]. Additionally, Bnip3 is essential for mitochondrial bioenergetics during adipocyte remodeling, and previous study has addressed that Bnip3 expression positively relates with adipose storage capacity [27]. Moreover, the latest studies have indicated that Bnip3 is a key effector of PPAR mediated adipose expression, and Bnip3$^{-/-}$ mice lead to systemic metabolic dysfunction [28]. Our findings suggest that cistanche promotes adipogenesis in 3T3-L1 cells, which is accompanied by up-regulated mRNA expression of Bnip3 and PPAR signaling pathway. Moreover, the Biosynthesis of secondary metabolites pathway and the Galactose metabolism pathway were primarily enriched in the top 20 significant DEGs pathways. These signaling pathways are closely related to the accumulation of lipids.

## Conclusion

In conclusion, we discovered that cistanche could promote the adipogenesis of 3T3-L1 preadipocytes. The genomic changes were analyzed using transcriptome sequencing technology. The transcriptome analysis revealed that the Differ and experimental groups had 14251 co-expressed genes. Between the Differ versus Boiled group, Differ versus Enzymed group, Differ versus Boiled with Enzymed group Differ vs Boiled group, 368, 11 and 44 differential genes were detected, respectively. And DEGs were mainly enriched in adipogenesis-related pathways such the insulin signaling pathway and AMP-activated protein kinase (AMPK) signaling

pathway. Moreover, in the cistanche-treated group, the mRNA levels of key transcription factors for adipogenesis such as PPAR, AP2, LPL and DEGs Bnip3 were up-regulated. These findings provide new insight into the mechanism by which cistanche promote the adipogenesis of 3T3-L1 preadipocyte and identify Bnip3 as a new molecular target for adipogenesis.

## Supporting information

**S1 Fig. The heatmap of important DEGs associated with adipogenesis.** DEGs were presented in heatmap. Red indicates high expression genes, while blue indicates low expression genes. Color changing from red to blue indicate that log 10 (FPKM+1) gradually changes from big to small. Nor: Normal group. Differ: 3T3-L1 cells were treated with differentiation medium. Boiled: 3T3-L1 cells were treated with cistanche of Boiled. Enzymed: 3T3-L1 cells were treated with cistanche of Enzymed. Boiled with Enzyme: 3T3-L1 cells were treated with cistanche of Boiled with enzymed.
(TIF)

**S2 Fig. The 3T3-L1 cell under optical microscope.** The figure of 3T3-L1 cells in pre-staining. (A) Normal group. (B) 3T3-L1 cells were treated with differentiation medium. (C) 3T3-L1 cells were treated with cistanche of Boiled. (D) 3T3-L1 cells were treated with cistanche of Enzymed. (E) 3T3-L1 cells were treated with cistanche of Boiled with Enzymed.
(TIF)

**S1 Graphical abstract.**
(TIF)

## Author Contributions

**Investigation:** Chen Zhao.

**Writing – original draft:** Ping Zhang, Feifan Ma.

**Writing – review & editing:** Le Su, Xiuyu Ji, Qiulin Yue, Song Zhang, Xin Sun, Kunlun Li, Lin Zhao.

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
