## [Decision Letter · Decision Letter 0]

14 Oct 2021

PONE-D-21-25814Cistanche promotes the adipogenesis of 3T3-L1 preadipocytesPLOS ONE

Dear Dr. Zhao,

Thank you for submitting your manuscript to PLOS ONE. After careful consideration, we feel that it has merit but does not fully meet PLOS ONE’s publication criteria as it currently stands. Therefore, we invite you to submit a revised version of the manuscript that addresses the points raised during the review process. I feel that in this manuscript the results are not sufficeinet to publish in the current form. There are not any mRNA level of key trancription factors for adipogenesis such as PPAR aP2, LPL, were shown with any treatment only Nile Red staining and Oil Red O staining were shown in the result. Also there are not any validation of RNA-seq data by qPCR which should be included in the results.

I am including the comments that reviewer made on your paper, which I hope you will find useful and constructive. As you will see, the reviewer expresses their view in the study, and they have a number of criticisms and suggestions. They are all suggesting changes to strengthen this aspect of the study. Based on these comments, it seems premature to proceed with the paper in its current form; however, if it is possible to address the concerns raised with additional data and/or discussion, we would be interested in considering a revised version of the manuscript.

We look forward to receiving your revised manuscript.

Kind regards,

Abdul Qadir Syed, PhD

Academic Editor

PLOS ONE

Journal Requirements:

2. We note that you are reporting an analysis of a microarray, next-generation sequencing, or deep sequencing data set. PLOS requires that authors comply with field-specific standards for preparation, recording, and deposition of data in repositories appropriate to their field. Please upload these data to a stable, public repository (such as ArrayExpress, Gene Expression Omnibus (GEO), DNA Data Bank of Japan (DDBJ), NCBI GenBank, NCBI Sequence Read Archive, or EMBL Nucleotide Sequence Database (ENA)). In your revised cover letter, please provide the relevant accession numbers that may be used to access these data. For a full list of recommended repositories, see http://journals.plos.org/plosone/s/data-availability#loc-omics or http://journals.plos.org/plosone/s/data-availability#loc-sequencing

We note that one or more of the authors is affiliated with the funding organization, indicating the funder may have had some role in the design, data collection, analysis or preparation of your manuscript for publication; in other words, the funder played an indirect role through the participation of the co-authors. If the funding organization did not play a role in the study design, data collection and analysis, decision to publish, or preparation of the manuscript and only provided financial support in the form of authors' salaries and/or research materials, please do the following:

1. Review your statements relating to the author contributions, and ensure you have specifically and accurately indicated the role(s) that these authors had in your study. These amendments should be made in the online form.

2. Confirm in your cover letter that you agree with the following statement, and we will change the online submission form on your behalf: 

“The funder provided support in the form of salaries for authors [insert relevant initials], but did not have any additional role in the study design, data collection and analysis, decision to publish, or preparation of the manuscript. The specific roles of these authors are articulated in the ‘author contributions’ section

Reviewers' comments:

Reviewer's Responses to Questions

**Comments to the Author**

1. Is the manuscript technically sound, and do the data support the conclusions?

Reviewer #1: No

2. Has the statistical analysis been performed appropriately and rigorously? 

Reviewer #1: Yes

3. Have the authors made all data underlying the findings in their manuscript fully available?

Reviewer #1: Yes

4. Is the manuscript presented in an intelligible fashion and written in standard English?

Reviewer #1: No

5. Review Comments to the Author

Reviewer #1: The authors have shown the effect of Cistanche on adipogenesis. Although the work is novel, the experimental details, the representation of results and the discussion don’t meet the publication criteria as of now. However, the authors can consider revising and improve the manuscript.

Major comments

1.The English needs to be improved throughout the text. The figure numbers need to be uniform. The figure legends should be detailed and explanatory. Example Figure 2A should mention the details of all the stains used. Figure 3 showing RNA sequencing result should be detailed.

2. Figure a in both Figure 2 A and 2B representing the normal group shows very less cell numbers as compared to the treated group. As the cell number is different in each group, the quantification of lipid accumulation in terms of absorbance will not confirm the increase in lipid content. Either fixed cell number or amount of protein needs to be considered during final calculation.

Further, figure legend or material section doesn’t show the time of incubation in different treatments.

3. Both RNA sequencing results along with the Enrichment analysis needs to be explained properly in the result section.

5. As the authors mentioned about the expression of Ghrp, Bnip3 and Gipr genes were significantly increased, representation of selected genes with a heatmap can emphasize the difference.

4. Authors should rewrite the discussion while properly discussing the outcomes and the significance of the outcomes. The pathways which significantly changed with the treatment of Cistanche needs to be emphasized properly.

The work in the manuscript is novel and contains a large amount of data from RNA sequencing. However, the data representation can be organized and improved in both result and discussion section to show the novelty and importance of the work.

6. PLOS authors have the option to publish the peer review history of their article (what does this mean?). If published, this will include your full peer review and any attached files.

Reviewer #1: No

---

## [Author Response · Author response to Decision Letter 0]

30 Dec 2021

Dear Editor and Reviewers,

Thank you for your letter and for the reviewers’ comments concerning our manuscript entitled “Cistanche promotes the adipogenesis of 3T3-L1 preadipocytes” (PONE-D-21-25814). The comments are all valuable and very helpful for revising and improving our paper, as well as the important guiding significance to our researches. We have read the comments carefully and made corrections which we hope meet with approval. Revised portions are marked in shadow in our revised manuscript. 

Editor comments：

1. There are not any mRNA level of key trancription factors for adipogenesis such as PPAR aP2, LPL, were shown with any treatment only Nile Red staining and Oil Red O staining were shown in the result. Also there are not any validation of RNA-seq data by qPCR which should be included in the results.

We appreciate constructive comments from the associate editor. We have examined the mRNA level of key transcription factors for adipogenesis such as PPAR, AP2 and LPL (Fig 8). Additionally, to validate changes in gene expression patterns, we identified and verified the expression level of Bnip3, a gene related to adiopogenesis by using RT-PCR. The results indicated that cistanche promoted adipogenesis, which was accompanied by up-regulated level of Bnip3 and PPAR in Boiled group. We tried our best to improve the manuscript and made some changes in the manuscript. These changes will not influence the content and framework of the paper. And here we did not list the changes but marked in shadow in our revised manuscript. We hope that the correction will meet with approval. Once again, thank you very much for your comments and suggestions.

Reviewer Comments:

1. The English needs to be improved throughout the text. The figure numbers need to be uniform. The figure legends should be detailed and explanatory. Example Figure 2A should mention the details of all the stains used. Figure 3 showing RNA sequencing result should be detailed.

Response: The English has been improved throughout the text. We really hope that the flow and language level have been substantially improved. The figure numbers are uniformed and the figure legends are enriched in revised manuscript. Additionally, Fig 2A was the Oil Red O staining of differentiated 3T3-L1 adipocytes. The lipid drops were stained in red and nucleus was stained in blue, and the more red dots in the figure, the more lipid accumulation. Fig 3 was the Pearson correlation coefficients of all gene expression. Pearson correlation coefficients of all gene expressions between each two samples were compared. The higher the correlation coefficient was, the more similar the gene expression level was. And the correlation coefficient values above 0.995 are presented as red and yellow, the correlation coefficient values below 0.995 are indicated in blue. These samples were highly correlated and did not have larger difference in the same group, the cistanche experimental group were separated from control. These figure legends were enriched in the revised manuscript.

2. Figure are in both Figure 2 A and 2B representing the normal group shows very less cell numbers as compared to the treated group. As the cell number is different in each group, the quantification of lipid accumulation in terms of absorbance will not confirm the increase in lipid content. Either fixed cell number or amount of protein needs to be considered during final calculation. Further, figure legend or material section doesn’t show the time of incubation in different treatments.

Response: Thank you for your careful review and constructive suggestion regarding our manuscript. Firstly, the number of cells is uniformed in a 6-well culture plate at a density of 1×105 cells/ml in normal and treated group following previously established standard culture system [1]. Additionally, the number of cells in the Differ group is appropriate, and the comparison between the experimental group and Differ control group will be more reasonable. Moreover, Oil red O and Nile red stainings were performed to exhibit the amount of lipid droplets rather than the cell number, The triglyceride, lipoprotein and lipid of intracellular could be stained into jacinth and red by ORO and Nile red staining. Thus, the quantification of triglyceride, lipoprotein and lipid are included in terms of absorbance in final calculation. We have enriched the figure of pre-staining cells in supplement materials, which could present the cell status and number (S2 Fig). As for the time of incubation, two days after cells reached confluence, cell differentiation was then induced with 10% FBS DMEM contain MDI (0.5μM isobutylmethylxanthine IBMX, 5μM dexamethasone and 0.5 μg/ml insulin) with cistanche for 2 days. Next, the medium was replaced with 10% FBS DMEM containing 5 μg/ml insulin with cistanche for 2 days. Next, the differentiation medium was replaced with 10% FBS DMEM contains cistanche for another 2 days. We added this information in the material section in the revised manuscript.

3. Both RNA sequencing results along with the Enrichment analysis needs to be explained properly in the result section.

Response: We have found this suggestion very useful to confirm the experimental results. However, the analysis results we got from the sequencing HuaDa Gene Company is about the differential genes (DEGs) of the experimental groups. The company did not provide us the Enrichment data. Moreover, in many research articles, differential gene RNA transcriptome sequencing is mainly used to analyze DEGs in different groups [2, 3]. Thus, the Enrichment analysis could not be provided in the manuscript.

4. As the authors mentioned about the expression of Ghrp, Bnip3 and Gipr genes were significantly increased, representation of selected genes with a heatmap can emphasize the difference.

Response: We thank the reviewer for this comment. We have re-analysis the data among these DEGs, and the important DEGs were presented in heatmap (S1 Fig). We re-analysed the comparison between Differ and cistanche treated groups. The level of Bnip3 was significantly increased in all comparison of Differ and cistanche treated group. To validate changes in gene expression patterns, we identified and examined the expression level of Bnip3, a gene related to adiopogenesis by using RT-qPCR. The data showed that the level of Bnip3 was significantly increased in Boiled group. Additionally, the results of key transcription factors for adipogenesis such as PPAR, aP2 and LPL were measured. The results indicated that cistanche promoted adipogenesis, which was accompanied by up-regulated levels of Bnip3 and PPAR in Boiled group. We added these results in the revised manuscript.

5. Authors should rewrite the discussion while properly discussing the outcomes and the significance of the outcomes. The pathways which significantly changed with the treatment of Cistanche needs to be emphasized properly.

Response: We have enriched the outcomes and the significance of the outcomes in the discussions in the revised manuscript. Cistanche is a traditional herb with a wide range of medicinal properties. Previous studies has showed that cistanche has a wide range of possible effects on lipid peroxidation and lipid accumulation, but its regulatory effects and mechanism on 3T3-L1 adipocytes are unknown. We expounded the mechanism of cistanche promoted the adipogenesis from the droplet formation and genetic level, which may help to bridge this gap. Moreover, in the cistanche-treated group, the mRNA levels of key transcription factors for adipogenesis such as PPAR, AP2, LPL and DEGs Bnip3 were up-regulated. These findings provide new insight into the mechanism by which cistanche promote the adipogenesis of 3T3-L1 preadipocyte and identify Bnip3 as a new molecular target for adipogenesis.

In additionally, our results showed that the mRNA levels of Bnip3 were up-regulated in cistanche treated group. Studies have reported that Bnip3 encodes the mitophagy signaling pathway, which is associated with the lipid metabolism in the liver [4]. Additionally, Bnip3 is essential for mitochondrial bioenergetics during adipocyte remodeling and relates with adipose storage capacity [5]. Moreover, the latest studies have indicated that Bnip3 is a key effector of PPAR mediated adipose expression, and Bnip3-/- mice lead to systemic metabolic dysfunction [6]. Our findings suggested that cistanche promotes adipogenesis in 3T3-L1 cells, which was accompanied by up-regulated mRNA expression of Bnip3 and PPAR signaling pathway. Moreover, the Biosynthesis of secondary metabolites pathway and the Galactose metabolism pathway were primarily enriched in the top 20 significant DEGs pathways. These signaling pathways are closely related to the accumulation of lipids. We have added these discussions in the revised manuscript.

References

1. Han JH, Jang KW, Park MH, Myung CS. Garcinia cambogia suppresses adipogenesis in 3T3-L1 cells by inhibiting p90RSK and Stat3 activation during mitotic clonal expansion. Journal of cellular physiology. 2021;236(3):1822-39. Epub 2020/07/28. doi: 10.1002/jcp.29964. PubMed PMID: 32716094.

2. Xin Y, Li C, Guo Y, Xiao R, Zhang H, Zhou G. RNA-Seq analysis reveals a negative role of MSMO1 with a synergized NSDHL expression during adipogenesis of 3T3-L1. Biosci Biotechnol Biochem. 2019;83(4):641-52. Epub 2018/12/26. doi: 10.1080/09168451.2018.1559719. PubMed PMID: 30582412.

3. Yun J, Jin H, Cao Y, Zhang L, Zhao Y, Jin X, et al. RNA-Seq Analysis Reveals a Positive Role of HTR2A in Adipogenesis in Yan Yellow Cattle. International journal of molecular sciences. 2018;19(6). Epub 2018/06/15. doi: 10.3390/ijms19061760. PubMed PMID: 29899319; PubMed Central PMCID: PMCPMC6032390.

4. Zhang Y, Wang Y, Wang X, Ji Y, Cheng S, Wang M, et al. Acetyl-coenzyme A acyltransferase 2 promote the differentiation of sheep precursor adipocytes into adipocytes. Journal of cellular biochemistry. 2018. Epub 2018/11/30. doi: 10.1002/jcb.28080. PubMed PMID: 30485515.

5. Choi JW, Jo A, Kim M, Park HS, Chung SS, Kang S, et al. BNIP3 is essential for mitochondrial bioenergetics during adipocyte remodelling in mice. Diabetologia. 2016;59(3):571-81. Epub 2015/12/24. doi: 10.1007/s00125-015-3836-9. PubMed PMID: 26693709.

6. Tol MJ, Ottenhoff R, van Eijk M, Zelcer N, Aten J, Houten SM, et al. A PPARγ-Bnip3 Axis Couples Adipose Mitochondrial Fusion-Fission Balance to Systemic Insulin Sensitivity. Diabetes. 2016;65(9):2591-605. Epub 2016/06/22. doi: 10.2337/db16-0243. PubMed PMID: 27325287; PubMed Central PMCID: PMCPMC5001173.

---

## [Decision Letter · Decision Letter 1]

17 Feb 2022

Cistanche promotes the adipogenesis of 3T3-L1 preadipocytes

PONE-D-21-25814R1

Dear Dr. Zhao,

We’re pleased to inform you that your manuscript has been judged scientifically suitable for publication and will be formally accepted for publication once it meets all outstanding technical requirements but I highly recommend to provide better microscopic images for the final publication which also a reviewer concern.

Kind regards,

Abdul Qadir Syed, PhD

Academic Editor

PLOS ONE

Additional Editor Comments (optional):

Reviewers' comments:

Reviewer's Responses to Questions

**Comments to the Author**

1. If the authors have adequately addressed your comments raised in a previous round of review and you feel that this manuscript is now acceptable for publication, you may indicate that here to bypass the “Comments to the Author” section, enter your conflict of interest statement in the “Confidential to Editor” section, and submit your "Accept" recommendation.

Reviewer #1: All comments have been addressed

2. Is the manuscript technically sound, and do the data support the conclusions?

Reviewer #1: Yes

3. Has the statistical analysis been performed appropriately and rigorously? 

Reviewer #1: Yes

4. Have the authors made all data underlying the findings in their manuscript fully available?

Reviewer #1: Yes

5. Is the manuscript presented in an intelligible fashion and written in standard English?

Reviewer #1: Yes

6. Review Comments to the Author

Reviewer #1: Authors have answered all the comments raised. High Res microscopic images can be provided for better visualization.

7. PLOS authors have the option to publish the peer review history of their article (what does this mean?). If published, this will include your full peer review and any attached files.

Reviewer #1: No

---

## [Editor Report · Acceptance letter]

21 Feb 2022

PONE-D-21-25814R1 

Cistanche promotes the adipogenesis of 3T3-L1 preadipocytes 

Dear Dr. Zhao:

I'm pleased to inform you that your manuscript has been deemed suitable for publication in PLOS ONE. Congratulations! Your manuscript is now with our production department. 

Kind regards, 

on behalf of

Dr. Abdul Qadir Syed 

Academic Editor

PLOS ONE